# Cranial Electrode Belt Position Improves Diagnostic Possibilities of Electrical Impedance Tomography during Laparoscopic Surgery with Capnoperitoneum

**DOI:** 10.3390/s23208644

**Published:** 2023-10-23

**Authors:** Kristyna Koldova, Ales Rara, Martin Muller, Tomas Tyll, Karel Roubik

**Affiliations:** 1Department of Biomedical Technology, Faculty of Biomedical Engineering, Czech Technical University in Prague, 272 01 Kladno, Czech Republic; kristyna.koldova@fbmi.cvut.cz; 2Department of Anesthesiology, Resuscitation and Intensive Care Medicine First Faculty of Medicine, The Military University Hospital Prague, Charles University, 121 08 Prague, Czech Republic; ales.rara@uvn.cz (A.R.); tomas.tyll@uvn.cz (T.T.); 3Clinic of Anesthesiology, Critical Care 1st Faculty of Medicine, Thomayer University Hospital Prague, Charles University, 140 59 Prague, Czech Republic; martin.muller@ftn.cz

**Keywords:** electrical impedance tomography, belt position, laparoscopy, capnoperitoneum

## Abstract

Laparoscopic surgery with capnoperitoneum brings many advantages to patients, but also emphasizes the negative impact of anesthesia and mechanical ventilation on the lungs. Even though many studies use electrical impedance tomography (EIT) for lung monitoring during these surgeries, it is not clear what the best position of the electrode belt on the patient’s thorax is, considering the cranial shift of the diaphragm. We monitored 16 patients undergoing a laparoscopic surgery with capnoperitoneum using EIT with two independent electrode belts at different tomographic levels: in the standard position of the 4th–6th intercostal space, as recommended by the manufacturer, and in a more cranial position at the level of the axilla. Functional residual capacity (FRC) was measured, and a recruitment maneuver was performed at the end of the procedure by raising the positive end-expiratory pressure (PEEP) by 5 cmH_2_O. The results based on the spectral analysis of the EIT signal show that the ventilation-related impedance changes are not detectable by the belt in the standard position. In general, the cranial belt position might be more suitable for the lung monitoring during the capnoperitoneum since the ventilation signal remains dominant in the obtained impedance waveform. FRC was significantly decreased by the capnoperitoneum and remained lower also after desufflation.

## 1. Introduction

Laparoscopy is a minimally invasive method of abdominal surgery that is increasingly used in clinical practice. To create a proper approach to the needed tissue, a gas is insufflated into the peritoneal cavity and a pneumoperitoneum is created. Thanks to its blood solubility and non-toxic characteristics, the most frequently used gas for this purpose is carbon dioxide. Therefore, pneumoperitoneum is called capnoperitoneum. To ensure optimal working conditions in the peritoneal cavity, the pressure of 12–15 mmHg [1,2] is maintained during the laparoscopic surgery, which leads to several adverse effects. Studies show that capnoperitoneum leads to a cranial shift of the diaphragm [3,4] and a decrease of functional residual capacity (FRC), forced vital capacity (FVC) and a development of atelectasis and effusion [4,5,6].

To monitor and minimize these effects, there is an effort growing among researchers to use electrical impedance tomography (EIT) during laparoscopic surgeries with capnoperitoneum. EIT is a lung monitoring technique based on the injection of small electrical currents and voltage measurements using electrodes on the patient’s skin. Cross-sectional images are generated from these measurements, representing the impedance change in a 5- to 10-cm-wide slice of the thorax [7]. It is a radiation-free, noninvasive, portable technique imaging the distribution of ventilation. Several studies [8,9,10,11,12,13] used EIT to monitor the lungs during capnoperitoneum for the purpose of analyzing regional ventilation changes and the effect of positive end-expiratory pressure (PEEP) changes. They all used a standardly placed electrode belt as recommended by the manufacturers of the EIT systems. However, the methodology of EIT use for this application is unclear, since the standard electrode belt position for lung monitoring with the mostly used EIT system PulmoVista 500 (Dräger Medical, Lübeck, Germany) is in the 4th–6th intercostal space, where the diaphragm is present during laparoscopy, as a recent study [3] has proven. The manufacturer warns in their educational brochure [14] and in the instructions for use (IFU) [15] and several studies [7,16] prove that placing the electrode belt in the juxta-diaphragmatic position can severely influence the results by the diaphragm’s entering the EIT sensitivity region. However, the caudal thoracic level just above the diaphragm is of particular importance to be monitored because lung pathologies such as atelectasis can be expected predominantly in this area.

No study systematically analyzing the feasibility and reliability of using the EIT for the lung monitoring during laparoscopic surgeries with capnoperitoneum has been found. However, the knowledge of the methodology seems crucial to avoid any inaccuracies and wrong therapeutical decisions based on EIT images.

The aim of this study was to verify the applicability of the use of electrical impedance tomography during laparoscopic surgery with capnoperitoneum from both medical and technical points of view. Another aim was to analyze the extent of the ventilation component in the EIT signal obtained during laparoscopic surgery with capnoperitoneum using electrical impedance tomography with the electrode belt in two positions: in the standard 4th–6th intercostal space and in the cranial position at the level of the axilla, between the 2nd–4th intercostal space. In order to explain the differences caused by the EIT belt position and the presence of capnoperitoneum, the changes of the spectral components of the EIT signal caused by ventilation and perfusion were analyzed. Another aim of the study was therefore to evaluate the benefit of cranial position of the electrode belt used for lung monitoring with EIT during laparoscopic surgery with capnoperitoneum on the evaluation of distribution of ventilation.

## 2. Materials and Methods

### 2.1. Patients

A prospective interventional study took place in the Military University Hospital in Prague, Czech Republic. The study was approved by the local Ethics committee on 23 November 2015, with Reference Number 108/8-104/2015-UVN. The study was registered in ClinicalTrials.gov with an identifier NCT03038061. The study uses the crossover design. The study sample consisted of 16 patients (11 females, 5 males), average age 46.5 ± 15 years, average BMI 27 ± 5 kg/m^2^. All the patients enrolled in the study underwent laparoscopic surgery with capnoperitoneum (cholecystectomy, hernioplasty or gynecological procedures) and were in the supine position. The basic characteristics of the enrolled patients are summarized in Table 1. Exclusion criteria for the patient enrolment in the study were heart or lung diseases. Standard exclusion criteria for the EIT use as listed in the IFU [15] were applied as well. All patients participated voluntarily and signed an informed consent form prior to the enrolment in the study.

### 2.2. Anesthesia and Patient Care

All surgical procedures were performed under general anesthesia. Initiation and management of anesthesia were in the mode of total intravenous anesthesia (TIVA). A combination of propofol, sufentanil and non-depolarizing myorelaxants (rocuronium or cisatracurium) was administered. After induction of anesthesia, the patient was orotracheally intubated and connected to the ventilator of a Primus anesthesia machine (Dräger Medical, Lübeck, Germany). After the patient was adequately prepared (correct position of the endotracheal cannula and its fixation were verified), the patient was switched to the ventilator Engström Carestation (Datex-Ohmeda, GE Healthcare, Helsinki, Finland), which allowed the measurement of functional residual capacity (FRC) using the nitrogen washout technique with a step change in the concentration of supplied oxygen. The volume-controlled ventilation (VCV) mode of mechanical ventilation was used with tidal volumes of 6–8 mL/kg of ideal body weight, positive end-expiration pressure (PEEP) of 5–8 cmH_2_O, respiratory rate was managed according to capnography (end-tidal concentration of carbon dioxide, EtCO_2_) to sustain normocapnia (4–5 kPa), fraction of inspired oxygen (FiO_2_) was 50%. Consequently, the ventilatory parameters were set and adjusted individually according to the development of oxygen saturation and the assessment and evaluation of the anesthesiologist. The High-Flow Insufflation Unit UHI-3 (Olympus Surgical Technologies, Hamburg, Germany) was used for the insufflation and maintenance of the capnoperitoneum.

### 2.3. Measurements

Electrical impedance tomography system PulmoVista 500 (Dräger Medical, Lübeck, Germany) was used to monitor distribution of ventilation. This EIT system uses an electrode belt with 16 electrodes implemented in the belt, and one standard single use reference electrode. The feed current frequency is in the range of 80 to 130 kHz and is set automatically by the system in order to minimize noise in the image and electromagnetic interference from other devices. The feed current amplitude is 80 to 90% of the maximum patient auxiliary current conforming to the international standard [17]. The frame rate used was set automatically to 50 frames per second. The EIT system performs automatic calibration prior to the monitoring of every new patient. Skin contact was enhanced with electrode gel to ensure minimal contact resistance. Even though the EIT system requested calibration every time the electrode belts and patient’s cables from different tomographic planes were switched, the calibration was performed only once at the beginning of the measurement in order to keep the same conditions during the measurements and to assure valid offline data analysis.

### 2.4. Experimental Protocol

Two individual electrode belts were placed on a patient, one in the standard position at the 4th–6th intercostal space in the medio-clavicular line, as recommended by the manufacturer, and another one in a more cranial position at the level of the axilla, between the 2nd–4th intercostal space in the medio-clavicular line. Due to the mutual interference, it was not possible to perform the EIT monitoring from both levels simultaneously, so we switched the belts using only one EIT system and recorded from one tomographic plane at a time. Measurements from both tomographic planes were performed in the following five phases: (1) spontaneous breathing before general anesthesia induction, (2) mechanical ventilation before the beginning of the surgery, (3) mechanical ventilation during the insufflation of the capnoperitoneum and at the time of maintaining the capnoperitoneum, (4) mechanical ventilation after finishing the procedure; and (5) mechanical ventilation with raised value of the positive end-expiratory pressure (PEEP) by 5 cmH_2_O (moderate recruitment PEEP maneuver). Also, for the phases 2–5, values of FRC were measured. Using this parameter, we could analyze the effect of capnoperitoneum and PEEP and to corelate changes in FRC with changes in relative impedance. It was necessary to perform EIT measurements only at the time when the electrosurgical unit (ESU) was not in use, otherwise the device switched into the safe mode automatically and an error message “Safety function activated” occurred on the screen. Then, the EIT system had to be restarted, and the measurement had to be started again. Since the EIT measurement is not compatible with the ESU, and the EIT recording is terminated when the ESU is in use, it was necessary to communicate properly and closely cooperate with the surgeon. The scheme of the measurement is depicted in Figure 1.

### 2.5. EIT Data Analysis

The data analysis was performed using off-line software Dräger EIT Data Analysis Tool 6.1. Version 1.n (Dräger Medical, Lübeck, Germany), MATLAB R2022a (MathWorks, Natick, MA, USA), and Microsoft Excel 365 (Microsoft, Redmond, Washington, DC, USA). Based on the principle of the functional EIT measurements, the provided values of relative impedance are in Arbitrary Units (AU), since the EIT system does not calculate absolute values of impedance.

In the Dräger EIT Analysis Tool, the two standard ways of division into regions of interest (ROI) were selected: four horizontal layers and quadrants. The location of each ROI is shown in Figure 2. We also evaluated the effect of baseline change and lowpass filtration of the EIT signal. Two different baselines were used for EIT data reconstruction: the first baseline was pre-set automatically from the segment of EIT data being currently processed, and another baseline from the segment with mechanical ventilation was implemented manually into the segment with mechanical ventilation with capnoperitoneum. Also, the EIT data were compared with and without a low pass filter, with a cut-off frequency of 40/min, which was also set in the Dräger EIT Analysis Tool.

Using MATLAB, relative impedance waveforms of whole experiment were studied. For every patient and every phase, we performed Fast Fourier Transform (FFT) to analyze the change of the amplitude at the breathing frequency and compared the data from each tomographic plane from mechanical ventilation with mechanical ventilation with capnoperitoneum. For FFT, a 20s window from the analyzed impedance signal was used.

In Excel, we analyzed the values of tidal variations (TV) and their changes due to the capnoperitoneum and the distribution of ventilation in the regions of interest (ROI) both in quadrants and layers. Changes of FRC were analyzed and compared for each measured phase during mechanical ventilation (phase 2 of the experiment according to Figure 1), mechanical ventilation with capnoperitoneum (phase 3), mechanical ventilation after desufflation (phase 4) and mechanical ventilation with PEEP was raised by 5 cmH_2_O (5). FRC was also measured during the emergence from anesthesia after the PEEP maneuver, but, as expected, the measured values of FRC fluctuated greatly and cannot be used to evaluate the effect of the distension maneuver.

### 2.6. Statistical Analysis

Data are presented as mean value ± standard deviation (SD). Thanks to the crossover study design, no separate control group was needed; the standard position of the electrode belt was considered as the reference in this study. Therefore, the statistical analysis was performed using the two-tailed paired Student’s *t*-test. Normality of the data was tested using the Shapiro–Wilk test. *p* values were considered statistically significant when *p* < 0.05. Microsoft Excel 365 (Microsoft, Redmond, Washington, DC, USA) and Statistica 7.1 (StatSoft/TIBCO Software, Palo Alto, CA, USA) were used for the statistical analysis.

## 3. Results

For each patient, two sets of EIT data were recorded at both the standard and cranial positions during the surgery. The overall waveform of relative impedance for the whole experiment of one patient and for both tomographic planes is presented in Figure 3. An impedance waveform, such as the one presented, is a typical outcome that a physician would obtain simply from the EIT system. When the capnoperitoneum was established (Phase 3), the tidal variations disappeared from the EIT signal recorded at the standard position, but remained visible in the cranial position.

Lung images were also generated by the EIT system for each patient. Another example of a simple outcome from the EIT system is a lung image showing the distribution of ventilation. The images obtained during mechanical ventilation and capnoperitoneum for both standard and cranial planes are shown in Figure 4. The captured distribution of ventilation corresponds to the lungs in the maximal inspiration. Figure 4 shows the difference in the detected aeration of the lungs in the standard belt position and the cranial belt position during mechanical ventilation and mechanical ventilation with capnoperitoneum.

The changes in the frequency spectrum of the impedance signal for one of the patients are shown in Figure 5. We compared the frequency spectra of a signal from mechanical ventilation and from mechanical ventilation with capnoperitoneum in both EIT belt positions. The breathing activity had a frequency of 0.2 Hz (respiratory rate 12/min) and heart activity corresponded to 1.2–1.4 Hz (heart rate of 70–80/min). During mechanical ventilation, the breathing frequency was dominant in both tomographic planes. During mechanical ventilation with capnoperitoneum, the amplitude of the spectral component corresponding to the breathing frequency was suppressed and a signal with frequency of 1.2–1.4 Hz became dominant in the standard belt position. Only for three patients, the signal at breathing frequency remained dominant, but with significantly reduced amplitude. In the cranial belt position, there was also a detectable decrease in the amplitude at the breathing frequency, however, it remained dominant in the spectrum. The average decrease in the relative impedance amplitude for all patients during mechanical ventilation with capnoperitoneum compared to mechanical ventilation was by 71 ± 22% for the standard belt position, and by 33 ± 19% for the cranial belt position. The statistical difference for both planes was very significant (*** *p* < 0.001). The significant changes in the values of amplitude are also emphasized by very different scale on the y-axes for the standard belt position in Figure 5.

In the off-line software Dräger EIT Data Analysis Tool 6.1. (Dräger Medical, Lübeck, Germany), it is possible to set the filtration frequency of the signal and to set the baseline frame, which is used as a reference level to display the impedance changes and lung image. The statistical significance (paired *t*-test) of the changes in distribution of ventilation in each ROI comparing mechanical ventilation and mechanical ventilation with capnoperitoneum for both tomographic planes is summarized in Table 2. The data were at first compared without any filtration and with the baseline frame pre-set automatically (baseline from mechanical ventilation and baseline from mechanical ventilation with capnoperitoneum). Then, we used a low pass filter (cut-off frequency of 40/min) and a pre-set baseline. Then, no filter was used and one common baseline from mechanical ventilation was set manually also for the signal from mechanical ventilation with capnoperitoneum. Lastly, the data were compared after both the low pass filtration and setting of one common baseline from mechanical ventilation.

More of the significant changes were detected when comparing the data from the standard position of the belt: 18 cases were significantly different and 14 were not. However, the changes occurred also in the cranial position of the belt: 13 cases were significantly different and 19 were not. However, there was no obvious pattern detected, we cannot state that the changes were likely to occur in some of the ROIs more than in others.

The measured values of FRC for every patient in every described phase during the surgery are summarized in a graph in Figure 6. There was a very significant decrease of FRC during mechanical ventilation with capnoperitoneum and a slight decrease remained also after the desufflation of the capnoperitoneum. During the recruitment maneuver when the PEEP value was raised by 5 cmH_2_O, FRC reached approximately the same value as during mechanical ventilation prior to the insufflation of the capnoperitoneum. Measured values of FRC after the recruitment maneuver and during the emergence from anesthesia fluctuated greatly and cannot be used to evaluate the effect of the maneuver.

## 4. Discussion

The main finding of this study was that from a medical point of view, capnoperitoneum has a statistically significant impact on the EIT data obtained from a standard tomographic plane. Cranial shift of the lung tissue led to a significant decrease and even the disappearance of the impedance changes caused by the breathing activity at the level of 4th–6th intercostal space. Instead of impedance changes caused by the breathing, the heart activity was detected by the EIT system in the standard electrode belt position, meaning that the extent of ventilation component in the EIT signal was only minor. The EIT images and data obtained from this manufacturer-recommended belt position might not provide any information about lung ventilation and evaluation of these data could lead to potential errors and inaccurate conclusions about the patient’s medical condition.

Another important finding relates to the technical issue of the EIT system, since due to electromagnetic incompatibility, it is not possible to use EIT system and electrosurgical unit simultaneously at the same time, make it very demanding to obtain relevant EIT data during any surgery, not only laparoscopic.

When the electrode belt was placed more cranially at the level of axilla, the EIT system showed more accurate images and data with detectable impedance changes caused by breathing. The heart activity was detected at this level as well, but did not interfere with the ventilation signal as significantly as it did in the standard position. Therefore, the described change of the position of the electrode belt cranially represents a better approach. However, it may not sufficiently cover the juxta-diaphragm areas of the lungs, which are diagnostically most important. The methodology of the EIT use with the cranial electrode belt needs to be standardized and studied on a bigger patient sample, as it might be possible to use the electrode belt in this position to evaluate distribution of ventilation during laparoscopic surgeries with capnoperitoneum, but always with respect to the interindividual variability of the patients.

The impedance waveform in Figure 3 depicted the changes in the impedance as we switched the electrode belts in the different tomographic planes. When the peritoneal cavity was being inflated, respiratory excursions gradually disappeared in the standard belt position and the values of global tidal variation that were initially visible decreased. By the time the peritoneal cavity was fully inflated, no respiratory excursions were clearly detectable for majority (13) of the patients. For three patients, the respiratory excursions remained visible, but with strongly suppressed amplitude. It was still possible to detect the respiratory excursions in the cranial position, even though the amplitude was also reduced during the mechanical ventilation with capnoperitoneum. Also, the area of lungs, as shown in Figure 4, indicated that the capnoperitoneum led to a major cranial shift of the lungs and only parts of the inferior lobes of the lungs might have been detected by the EIT system in the standard belt position.

The amplitude of the relative impedance decreased due to the presence of the capnoperitoneum in both belt positions. A more significant decrease was evaluated in the standard position (71 ± 22%), compared to the decrease in the cranial position (33 ± 19%). This trend was also confirmed by the frequency analysis, as shown in Figure 5, where a significant decrease of the amplitude at the frequency of 0.2 Hz (12/min) was detected. At the standard position, heart frequency (approximately 1.3 Hz, 78/min) dominated the frequency spectrum during mechanical ventilation with capnoperitoneum. It is possible that during capnoperitoneum, the EIT system detected mainly the activity of the heart and the impedance changes caused by the pulsative blood flow. Therefore, we assumed that the images and data that are shown by the EIT from the standard belt position did not provide sufficient and correct information about the distribution of ventilation in the lungs.

When comparing the effect of filtration, the low pass filter led to less significant results in the standard position, but more in the cranial position. This was probably because the heart activity was a major detected signal during mechanical ventilation, but during mechanical ventilation with capnoperitoneum the breathing impedance changes became dominant in the cranial position. Another factor that might have been significant for the data evaluation was the selection of the baseline frame. The EIT software automatically selects the image with the lowest impedance value as a baseline frame and assigns the calculated relative impedance the value of 0 AU. This image is then used as a reference to display other images. It is also possible to select the baseline frame manually. Even though there are studies [18] proving only minor effect of the baseline frame on the distribution of ventilation, the current study showed that in this case the selection of the baseline frame can be very significant when we compared to EIT data with baseline selected from the mechanical ventilation phase without capnoperitoneum, and also for the case of mechanical ventilation with capnoperitoneum. A much greater impact of the baseline frame selection was observed for the standard belt position (ROI Q2, Q3, Q4, L2 and L3), as summarized in Table 2. We assumed that this was caused by the greater shift of the impedance level where the changes of impedance values appeared, as evident from Figure 3. However, we realized that the correctness and the informative value of the results in Table 2 might be debatable since the impedance changes detected by the electrode belt in the standard position do not appear to correspond with the breathing activity and the ventilation-related changes of impedance. However, the Dräger EIT Data Analysis Tool 6.1. software (Dräger Medical, Lübeck, Germany) provides the values of distribution of ventilation not only for respiratory signal, but for any uploaded EIT data with any course. For the cases where the breathing excursions were not detected in the EIT signal, there is no point in determining the distribution of ventilation in the ROIs. Moreover, it might contribute to the additional confusion in the subsequent data analysis.

The measured values of FRC were following the expectations. The increased intraabdominal pressure, caused by the capnoperitoneum, led to a very significant decrease (*p* < 0.001) of FRC values when comparing the values of FRC during mechanical ventilation and mechanical ventilation with capnoperitoneum. The literature also confirmed that capnoperitoneum leads to the decreased values of FRC [19,20]. However, we also managed to prove that a moderate recruitment maneuver with an increase of PEEP value by 5 cmH_2_O against the PEEP value used during the anesthesia and surgery that was performed after the CO_2_ desufflation led to a statistically significant improvement of FRC. However, it was not possible to evaluate the efficiency of the recruitment mauver after the PEEP value was decreased, since measured FRC values fluctuated greatly due to the patient emergence from the anesthesia.

Another finding of this study is the fact that the EIT system is significantly disturbed by the electrical surgical unit (ESU) that is used almost continuously during the surgical procedure, as these two devices are not electromagnetically compatible. Therefore, it is very hard to obtain the EIT data during the surgery when the ESU is being used by the surgeon and the EIT monitoring can be turned on only when the ESU is not being used. It demands a close communication and cooperation with the surgeons. Some studies do mention this severe technical limitation [9,11,12,21,22], but it is quite uncertain how other researchers dealt with this issue.

The mentioned studies [7,9,10,11,12,13,16,21,22] relate to the use of EIT during laparoscopic surgery with capnoperitoneum. The studies [21,22] used the cranial position of the electrode belt and following studies highlighted technical limitations, such as electromagnetic incompatibility with the ESU [9,11,12,21,22] and the presence of the diaphragm in the standard position of the electrode belt [7,16].

However, it is necessary to emphasize that none of the studies thoroughly evaluated the applicability and reliability of the EIT system for lung monitoring during laparoscopic surgery. Some studies encountered technical limitations, but they did not address why wrong data were being evaluated as a ventilation signal by the EIT system.

Our presented study is unique within this field because it delves into the roots of the problems, addressing why the application of EIT is limited in this context. It also highlights the need for a comprehensive study in this area, as many existing studies may present inaccurate results. This underscores the significance of the study.

Laparoscopic procedures are increasingly indicated for a wide range of patients, which is related to the proven positive effect on the healing of surgical wounds and patient recovery [23,24,25]. Another impulse is the rapid development of robotic and robot-assisted operatives in urology, surgery, and gynecology, where it is necessary to maintain the capnoperitoneum for many hours. The use of EIT for the perioperative monitoring of ventilation during general anesthesia with artificial lung ventilation could be indicated primarily in patients with a long operating time and at the same time difficult ventilation, i.e., patients with chronic lung disease (COPD) and obese patients. For clinical use, the following possible recommendations emerge from the study. To monitor artificial lung ventilation during surgical procedures with capnoperitoneum, it is convenient to place the electrode belt in the cranial position. On the contrary, in the event that it is desirable to use EIT during recovery from general anesthesia to optimize ventilation and monitor the airiness of the lungs, it is advisable to use the standard belt position at the level of the 4th–6th intercostal space. From a surgical point of view, patients with chronic diseases of the respiratory and circulatory system also benefit from laparoscopic and robotic surgery, for whom, however, in terms of circulatory strain and deterioration of ventilation functions due to capnoperitoneum and often also the operating position, the operation itself is risky due to limited functional reserves of the respiratory and circulatory system. In these patients, extended monitoring (and especially non-invasive such as EIT) during surgery is desirable and could significantly reduce the operative risk. EIT can help to optimize ventilation and circulation parameters, or it can help to effectively solve these complications and thus prevent the need for conversion from laparoscopic and robotic surgery to a classic operative technique.

EIT could be a very promising method for extended monitoring of lung ventilation and perfusion during laparoscopic procedures. Its advantage is non-invasiveness and the absence of ionizing radiation. Continued research is needed before EIT can be beneficially used in laparoscopic and robotic surgeries. Further studies are needed to investigate, in particular, the ability of EIT to effectively detect changes in ventilation parameters caused by capnoperitoneum, especially changes in FRC/EELV and ventilation distribution. It is also necessary to perform studies focused on the clinical effect of using EIT in laparoscopic and robotic operations, e.g., the effect of intraoperative EIT monitoring on the number of forced conversions to open surgery and other respiratory and cardiovascular complications.

The current study has several limitations. One of the limitations includes the absence of the visualization of the position of the diaphragm in the individual phases in specific patients. Therefore, it is not possible to determine with certainty whether the disappearance of the variability of ventilation impedance at the standard position of the electrode belt is caused by the entrance of capnoperitoneum into the area captured by the caudal belt, or whether the disappearance of respiratory variations occurs mainly due to the redistribution of ventilation cranially. Intraabdominal pressure was also not monitored during the study, neither was the current pressure of capnoperitoneum recorded. Intra-abdominal pressure could also be related to the amplitude of respiratory impedance variations. The influence of the operative position was also not evaluated. Intraabdominal or intraperitoneal pressure of capnoperitoneum and the position undoubtedly also influence the value of FRC.

Also, it is quite challenging to interpret the results obtained from the cranial position of the electrode belt, since there are currently no studies analyzing the lungs at this level, either in the physiological position of the lungs or during the capnoperitoneum. This is a subject to further investigation in another study.

Limitations of the study also include the relatively small number of subjects with a relatively large variance in BMI. Nevertheless, this number of subjects is sufficient to demonstrate the problem with EIT data recording and evaluation during capnoperitoneum and to find its cause in the EIT signal frequency spectrum when EIT is recorded in the standard EIT belt position.

Also, the impossibility to use both electrode belts simultaneously during the surgery because of mutual interference and the termination of the EIT recording caused by electrosurgical unit are limiting as well, but those are limitations of EIT as a method.

## 5. Conclusions

The main finding of this study was that lung monitoring during laparoscopic surgery with capnoperitoneum using the EIT system PulmoVista 500 (Dräger Medical, Lübeck, Germany) with an electrode belt used at the standard tomographic plane 4th–6th intercostal space, as recommended by the manufacturer, did not provide the desirable information about lung aeration and regional ventilation, as the EIT signal does not contain the spectral components related to ventilation. More cranial placement of the electrode belt might be a better approach to obtain clinically relevant information about the distribution of ventilation.

## Figures and Tables

**Figure 1 sensors-23-08644-f001:**
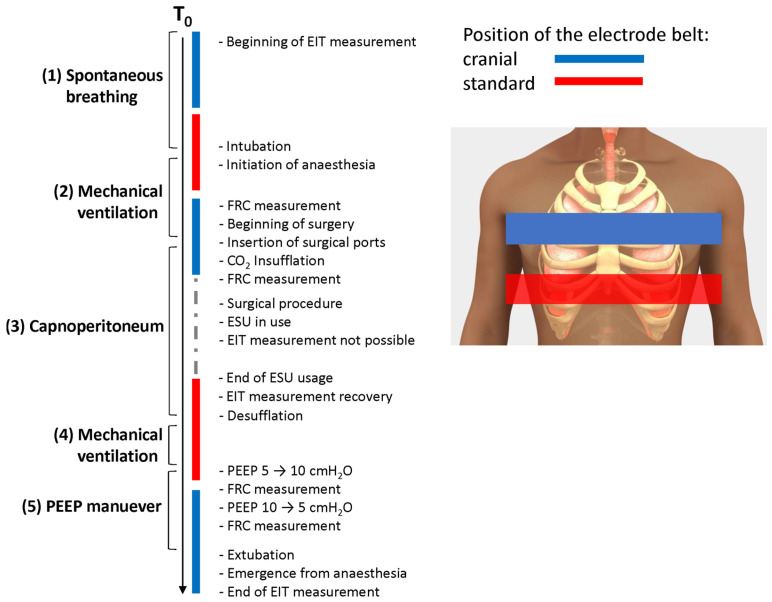
Schematic presentation of the timeline of the data collection. EIT was measured in two positions: cranial (blue) and standard (red) and the order of the belt selection was randomized.

**Figure 2 sensors-23-08644-f002:**
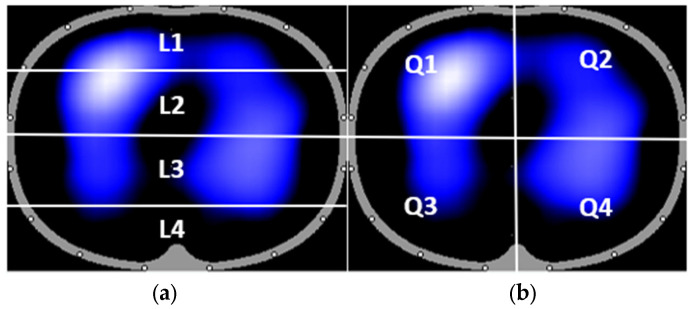
The standard division of EIT image into regions of interest (ROI): four horizontal layers L1–L4 (**a**) and quadrants Q1–Q4 (**b**).

**Figure 3 sensors-23-08644-f003:**
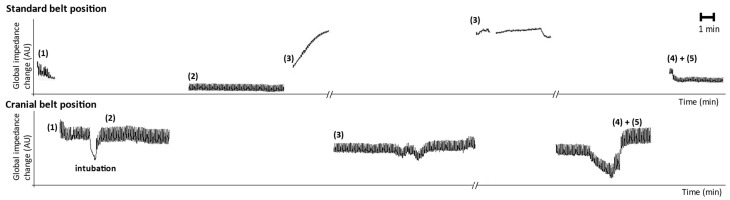
Global impedance waveform of the whole experiment for one patient. The upper waveform is recorded from the standard position of the electrode belt, the lower waveform from the cranial position of the electrode belt. Phases: (1) spontaneous breathing, (2) mechanical ventilation before the beginning of the surgery, (3) mechanical ventilation during the insufflation of capnoperitoneum and at the time of maintaining capnoperitoneum, (4) mechanical ventilation after finishing the procedure and (5) mechanical ventilation with raised value of positive end-expiratory pressure (PEEP) by 5 cmH_2_O. In order to emphasize the changes of impedance caused by the ventilation, data were filtered using low pass filter with a cut off frequency of 40/min.

**Figure 4 sensors-23-08644-f004:**
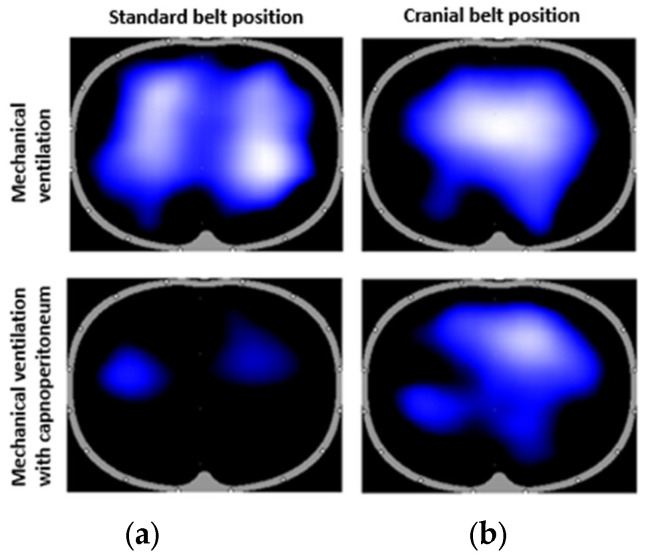
Images from the Dräger EIT Analysis Tool for one of the patients. The captured moment corresponds to the maximal inspiration during mechanical ventilation (top) and capnoperitoneum (bottom) in both standard (**a**) and cranial (**b**) tomographic planes.

**Figure 5 sensors-23-08644-f005:**
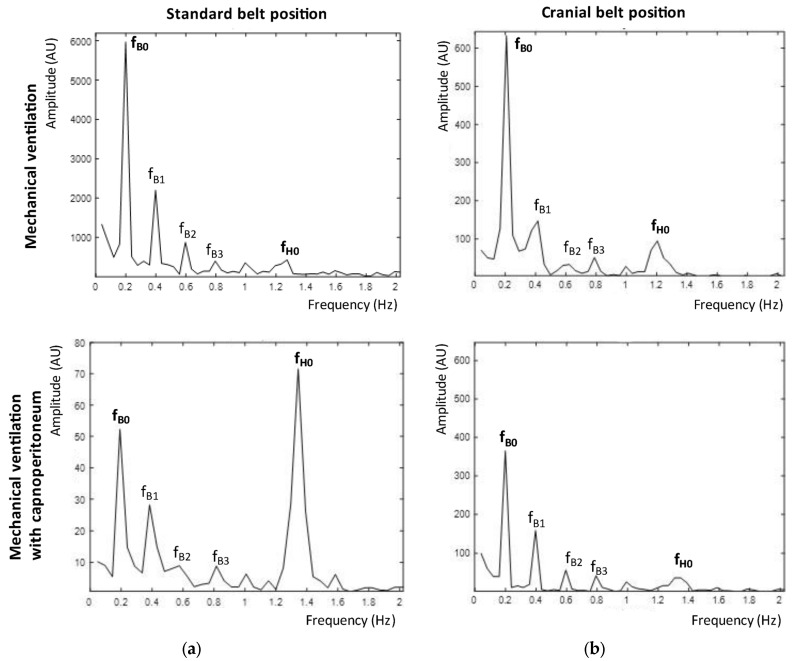
Frequency spectra of EIT recording for the standard plane (**a**) and the cranial plane (**b**) during mechanical ventilation (top) and during mechanical ventilation with capnoperitoneum (bottom). The breathing activity corresponded to a frequency of 0.2 Hz and is marked as a fundamental harmonic frequency f_B0_. Higher harmonic frequencies are marked as f_B1_, f_B2_, and f_B3._ Another detected fundamental harmonic frequency was a heart rate signal, which corresponded to a frequency of 1.2–1.4 Hz and is marked as f_H0_.

**Figure 6 sensors-23-08644-f006:**
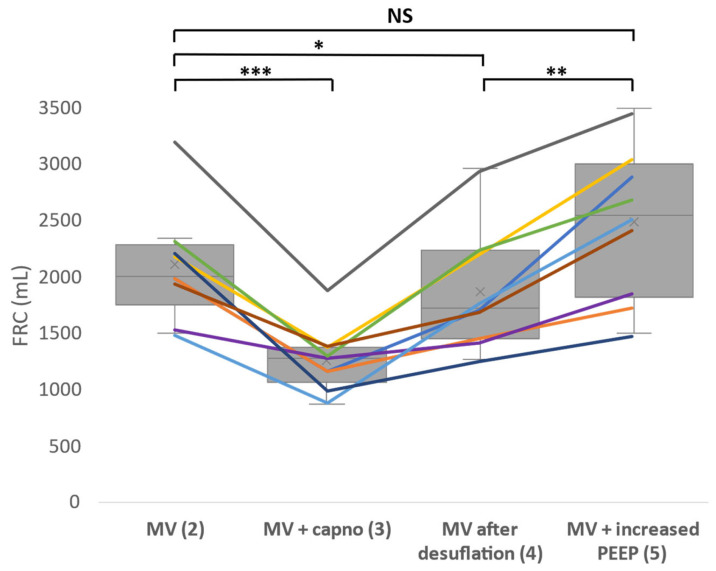
Functional residual capacity (FRC) changes during the whole experiment. Each line represents one patient. FRC was measured during mechanical ventilation (phase 2), mechanical ventilation with capnoperitoneum (phase 3), mechanical ventilation after desufflation (phase 4) and mechanical ventilation with PEEP raised by 5 cmH_2_O (phase 5). Statistical significance (paired *t*-test) is marked as follows: *** *p* < 0.001, ** *p* < 0.01, * *p* < 0.05 and NS as no statistical significance proven (*p* > 0.05). MV―mechanical ventilation.

**Table 1 sensors-23-08644-t001:** Characteristics of the patients involved in the study. Average values are presented as mean ± standard deviation (SD).

	Male/Female	Age (Years)	Weight (kg)	Height (cm)	BMI (kg/m^2^)	VT (mL)	Procedure
Patient 1	female	39	115	175	38	500	cholecystectomy
Patient 2	female	37	60	170	21	425	cholecystectomy
Patient 3	male	50	102	182	31	775	cholecystectomy
Patient 4	female	44	90	158	36	500	cholecystectomy
Patient 5	female	30	68	173	21	425	cholecystectomy
Patient 6	female	71	72	172	24	525	cholecystectomy
Patient 7	female	36	76	173	25	500	cholecystectomy
Patient 8	male	64	81	182	24	600	cholecystectomy
Patient 9	male	66	80	171	27	560	hernioplasty
Patient 10	female	25	82	173	28	500	ovarian cyst extirpation
Patient 11	female	41	72	168	25	500	ovarian cyst extirpation
Patient 12	female	51	82	185	24	400	hysterectomy
Patient 13	female	39	86	167	31	500	hysterectomy
Patient 14	male	59	98	184	29	525	hernioplasty
Patient 15	female	25	60	163	23	450	ovarian resection
Patient 16	male	67	75	176	24	490	hernioplasty
Summary	5/11	47 ± 15	81 ± 15	173 ± 7	30 ± 5	511 ± 87	

**Table 2 sensors-23-08644-t002:** Summary of the statistical significance of the changes in distribution of ventilation in each ROI when comparing mechanical ventilation and mechanical ventilation with capnoperitoneum for standard and cranial position of the electrode belt. Statistical significance is marked as follows: *** *p* < 0.001, ** *p* < 0.01, * *p* < 0.05 and NS as no statistical significance proven (*p* > 0.05).

	No Filter, Pre-Set Baseline		Low Pass Filter, Pre-Set Baseline		No Filter, Ventilation Baseline		Low Pass Filter, Ventilation Baseline	
ROI	Standard Position	Cranial Position	Standard Position	Cranial Position	Standard Position	Cranial Position	Standard Position	Cranial Position
Q1	*	*	NS	*	NS	NS	*	*
Q2	***	NS	*	NS	***	NS	**	*
Q3	*	NS	NS	NS	**	NS	NS	NS
Q4	NS	NS	*	*	**	NS	NS	*
L1	*	**	NS	*	NS	*	**	NS
L2	NS	*	*	**	*	NS	*	NS
L3	NS	NS	NS	*	**	NS	NS	*
L4	*	NS	NS	NS	NS	NS	*	NS

## Data Availability

The complete set of measured and evaluated raw EIT data is available online at: https://ventilation.fbmi.cvut.cz/data/ (accessed on 25 September 2023).

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
