# Peer review of "Cranial Electrode Belt Position Improves Diagnostic Possibilities of Electrical Impedance Tomography during Laparoscopic Surgery with Capnoperitoneum"

_sensors, 2023, doi:10.3390/s23208644_

Round 1

Reviewer 1 Report

The article titled "Cranial Electrode Belt Position Improves Diagnostic Possibilities of Electrical Impedance Tomography During Laparoscopic Surgery with Capnoperitoneum" presents interesting research on the benefits of using cranial electrode belt positions for electrical impedance tomography (EIT) during laparoscopic surgery with capnoperitoneum. While the study examines an important aspect of laparoscopic surgery, there are several areas that require improvement and clarification in order to strengthen the research.

1) Clear Objectives and Research Questions:
The article lacks a well-defined statement of objectives and research questions. It is crucial to clearly state the purpose of the study and the specific research questions that the investigation aims to answer. This will help readers understand the significance of the research and allow for a more focused discussion.

2) Sample Size and Generalizability:
The study's sample size is not clearly defined, and it is necessary to provide more information on the number of participants and their characteristics. Furthermore, the authors should consider including a broader range of participants to enhance the generalizability of the findings to a wider population. This will ensure that the results are applicable to a larger cohort of patients undergoing laparoscopic surgery with capnoperitoneum.

3) Methodological Rigor:
The article lacks a detailed description of the study design, including the specific steps taken during data collection, a clear explanation of the variables measured, and the methodology employed. It is essential to address these aspects in order to ensure the reproducibility of the study. Additionally, the statistical analysis conducted should be better explained, and the rationale behind the chosen statistical tests should be provided.

4) Lack of Comparison and Control Group:
The study does not mention a control or comparison group, making it challenging to evaluate the effectiveness of the cranial electrode belt position. The inclusion of a control group undergoing laparoscopic surgery without the cranial electrode belt would provide valuable insight and allow for a more comprehensive comparison. This would also help identify any confounding factors or variables affecting the results.

5) Discussion and Interpretation of Findings:
The discussion section of the article needs improvement. The authors should provide a detailed interpretation of the results, linking them back to the research questions and objectives. Additionally, the limitations of the study should be thoroughly discussed, addressing potential biases, confounders, and any limitations affecting the reliability and validity of the findings.

6) Practical Implications and Future Directions:
To enhance the relevance of the research, the article should include a section on the practical implications of the findings. It would be helpful to outline how the use of cranial electrode belt positions can improve laparoscopic surgery practices and patient outcomes. Furthermore, suggestions for future research directions and potential advancements based on the current study's limitations should be included.

Conclusion:
Overall, the article presents an intriguing study on the impact of cranial electrode belt positions during laparoscopic surgery, but there are several areas that require attention to strengthen the research. By addressing the issues mentioned above, the authors can improve the clarity, validity, and applicability of their findings to the medical community.

Reviewer 2 Report

General comment:

The manuscript introduces an experimental study on cranial electrode belt positioning to improve EIT-based diagnosis in laparoscopic surgeries. The work is relevant in the field of sensors and measurements with a clinical application. Furthermore, the proposal is well-motivated and represents an advance in its area of knowledge. The experimental framework is mostly clear. I have some points that should be addressed.

Comment 1:

In section 2, in the Measurements’ paragraph. It would be interesting to give more technical details on the measurement device, such as excitation frequency, current magntitude, electrodes’ features, etc..

Comment 2:

Regarding data analysis in section 2. Which was the length of the recorded data? Also, which was the topology of the low-pass filter? How did it is selected?

Comment 3:

It is interesting to see that the heart rate and respiratory frequency can be distinguished in the results. However, how does the proposal perform for retrieving respiration and perfusion signal? Does it is possible to give this valuable information?

Comment 4:

A comparison with similar works for this application is missing.

Round 2

Reviewer 1 Report

The article "Cranial electrode belt position improves diagnostic possibilities of electrical impedance tomography during laparoscopic surgery with capnoperitoneum" sheds light on a groundbreaking method that has the potential to revolutionize laparoscopic surgery. By introducing a cranial electrode belt position, the diagnostic capabilities of electrical impedance tomography (EIT) have been significantly enhanced. This article highlights the benefits and implications of this technique, paving the way for improved surgical outcomes and patient safety.
In conclusion, I belive the article is a valuable contribution to the field of laparoscopic surgery. The inclusion of the cranial electrode belt position significantly improves the diagnostic capabilities of EIT, thereby enhancing patient safety and surgical precision.